# A Penalized Likelihood Approach for Statistical Inference in a High-Dimensional Linear Model with Missing Data

**Jiwei Zhao** [1]

## Abstract

In this paper, we consider how to conduct statistical inference in a high-dimensional linear model where the response variable has missing values. Motivated by the fact that the missingness mechanism, albeit usually regarded as a nuisance, is largely unknown and difficult to specify, we adopt the conditional likelihood approach such that this nuisance can be completely ignored in our procedure. We establish the asymptotic theory of the proposed estimate and develop an easy-to-implement algorithm via some data manipulation strategy. Furthermore, we propose a data perturbation method for the variance estimation. The proposed methodology has broad potential for application in patient-reported outcomes or electronic health records. Although we do not have space to present our numerical results in this four-page extended abstract, we will definitely do so at the workshop if it is selected.

## 1. Introduction

A major step to achieve scientific discovery is to identify useful associations among different features and to quantify their uncertainties. This usually warrants building a regression model between an outcome variable and a set of covariates and estimating the coefficient as well as the precision of the estimate. Besides the traditional low-dimensional setting with a fixed and much smaller (than the sample size) dimensionality of the parameter, the modern high-dimensional regression usually posits a sparse parameter of interest and the well-studied regularization technique is frequently used to recover the sparsity.

In this paper, we analyze the high-dimensional setting, although the primary challenging difficulty we aim to address is the missing data issue, an inevitable concern in various disciplines ranging from biomedicine to social science. In the literature, the validity of a statistical method devised for missing data usually heavily depends on the assumption postulated on the so-called missingness mechanism (Little & Rubin, 2002); however, those assumptions are hard, if not infeasible, to be empirically verified. Due to the fact that the missingness mechanism is largely unknown and practically difficult to specify in applications, and that the occurrence of missing data is usually not the investigator's primary interest but complicates the statistical analysis, it is sensible to mainly study the regression model for the outcome while regarding the mechanism model as a nuisance and imposing a flexible assumption at the minimum level such that the protection to its model misspecification can be attained at the maximum level.

Therefore, we adopt a semiparametric framework consisting of a parametric regression, e.g., a linear model, for the outcome where the statistical task is to estimate the unknown parameter, perform variable selection and conduct post-selection inference, and a nonparametric and unknown nuisance for the missingness mechanism. Yet, this framework is not readily identifiable. In the past few years, statisticians have made great effort to advance the study of model identification by introducing a so-called instrument. The instrument could be a shadow variable (Shao & Zhao, 2013; Wang et al., 2014; Zhao & Shao, 2015; Miao & Tchetgen Tchetgen, 2016; Zhao & Ma, 2018; Miao et al., 2019) or an instrumental variable (Tchetgen Tchetgen & Wirth, 2017; Sun et al., 2018). Both approaches are reasonable and are suitable for different applications. We adopt the shadow variable approach in this paper primarily by virtue of the transparency of the regression model for the outcome. The details of the shadow variable will be presented in Section 2, but to be concise here, the shadow variable $Z$ is simply one component of the covariate $\mathbf{X}$: it could be multi-dimensional but we only discuss the one-dimensional case for the sake of maximum flexibility of the missingness mechanism. Based upon the existence of the shadow variable, we consider a conditional likelihood approach which will result in a nuisance-free procedure for estimation as well as for statistical inference.

---

[1]Department of Biostatistics, State University of New York at Buffalo, Buffalo, New York, USA. Correspondence to: Jiwei Zhao <jiwei2012zhao@gmail.com>.

*Proceedings of the $37^{th}$ International Conference on Machine Learning*, Vienna, Austria, PMLR 108, 2020. Copyright 2020 by the author(s).

Besides, there are two other features that are worth mentioning. The first is about our algorithm and computation. Although the conditional likelihood function we use looks complicated at first sight, through some data manipulation strategy, it can be analytically written as the likelihood of a conventional logistic regression with some prespecified format. Hence, our objective function can be readily optimized by many existing software packages; this greatly alleviates the computational burden of our procedure. Second, we provide both parameter estimation and variance estimation under high-dimensionality where we establish the so-called oracle property of the parameter estimate and also provide an easy-to-implement data-driven method for the variance estimation via a data perturbation technique.

## 2. Methodology

Denote the outcome variable $Y$ and covariate $\mathbf{X}$. We assume $\mathbf{X} = (\mathbf{U}^{\mathrm{T}}, Z)^{\mathrm{T}}$ where $\mathbf{U}$ is $p$-dimensional and $Z$ is univariate, with detailed interpretation later. In this paper, we consider the linear model

$$Y = \alpha + \boldsymbol{\beta}^{\mathrm{T}}\mathbf{U} + \gamma Z + \epsilon, \tag{1}$$

where $\boldsymbol{\beta}$ is also $p$-dimensional, $\alpha$ and $\gamma$ are scalars and $\gamma \neq 0$, $\epsilon \sim N(0, \sigma^2)$. We consider the situation that $Y$ has missing values while $\mathbf{X}$ is fully observed. We introduce a binary variable $R$ to indicate missingness: $R = 1$ if $Y$ is observed and $R = 0$ if missing. To allow the greatest flexibility of the missingness mechanism model, we assume

$$\mathrm{pr}(R = 1 \mid Y, \mathbf{X}) = \mathrm{pr}(R = 1 \mid Y, \mathbf{U}) = s(Y, \mathbf{U}), \tag{2}$$

where $s(\cdot)$ merely represents an unknown and unspecified function not depending on $Z$. We reiterate that, since the assumption (2), in a nonparametric fashion, does not specify a concrete form of $s(\cdot)$, one does not need to be worrisome of the mechanism model misspecification. Also, since it allows the dependence on $Y$, besides missing-completely-at-random (MCAR) and many scenarios of missing-at-random (MAR), the assumption (2) also contains various situations of missing-not-at-random (MNAR).

We term $Z$ the shadow variable following the work of (Miao & Tchetgen Tchetgen, 2016; Zhao & Ma, 2018; Miao et al., 2019; Zhao & Ma, 2019). Its existence depends on whether it is sensible that $Z$ and $R$ are conditionally independent (given $Y$ and $\mathbf{U}$) and that $Y$ heavily relies on $Z$ ($\gamma \neq 0$). There are many examples in the literature documenting that the existence of $Z$ is clinically sensible. Practically, a surrogate or a proxy of the outcome variable $Y$, which should not concurrently affect the missingness mechanism, might be a good choice for the shadow variable $Z$.

We assume independent and identically distributed observations $\{r_i, y_i, \mathbf{u}_i, z_i\}$ for $i = 1, ..., N$ and the first $n$ subjects

are free of missing data. Now we present a $s(y, \mathbf{u})$-free procedure via the use of the conditional likelihood.

Denote $\mathbf{V} = (Y, \mathbf{U}^{\mathrm{T}})^{\mathrm{T}}$. We start with

$$\prod_{i=1}^{n} p(\mathbf{v}_i \mid z_i, r_i = 1) = \prod_{i=1}^{n} \frac{s(\mathbf{v}_i)}{g(z_i)} p(\mathbf{v}_i \mid z_i), \tag{3}$$

where $g(z_i) = \mathrm{pr}(r_i = 1 \mid z_i) = \int \mathrm{pr}(r_i = 1 \mid \mathbf{v}) p(\mathbf{v} \mid z_i) d\mathbf{v}$ and $p(\cdot \mid \cdot)$ is a generic notation for conditional probability density/mass function. If $\mathbf{V}$ were univariate, we denote $\mathcal{R}$ as the rank statistic of $\{v_1, ..., v_n\}$, then

$$
\begin{aligned}
&\prod_{i=1}^{n} p(v_i \mid z_i, r_i = 1) \\
=\ & p(v_1, ..., v_n \mid z_1, ..., z_n, r_1 = \cdots = r_n = 1) \\
=\ & p(v_{(1)}, ..., v_{(n)}, \mathcal{R} \mid z_1, ..., z_n, r_1 = \cdots = r_n = 1) \\
=\ & p(\mathcal{R} \mid v_{(1)}, ..., v_{(n)}, z_1, ..., z_n, r_1 = \cdots = r_n = 1) \\
&\times p(v_{(1)}, ..., v_{(n)} \mid z_1, ..., z_n, r_1 = \cdots = r_n = 1)
\end{aligned}
\tag{4}
$$

The conditional likelihood that we use, the first term on the right hand side of (4), is exactly

$$
\begin{aligned}
&p(\mathcal{R} \mid v_{(1)}, ..., v_{(n)}, z_1, ..., z_n, r_1 = \cdots = r_n = 1) \\
=\ & \frac{p(v_1, ..., v_n \mid z_1, ..., z_n, r_1 = \cdots = r_n = 1)}{p(v_{(1)}, ..., v_{(n)} \mid z_1, ..., z_n, r_1 = \cdots = r_n = 1)} \\
=\ & \frac{\prod_{i=1}^{n} p(v_i \mid z_i, r_i = 1)}{\Sigma_{\omega \in \Omega} \prod_{i=1}^{n} p(v_{\omega(i)} \mid z_i, r_i = 1)} \\
=\ & \frac{\prod_{i=1}^{n} p(v_i \mid z_i)}{\Sigma_{\omega \in \Omega} \prod_{i=1}^{n} p(v_{\omega(i)} \mid z_i)},
\end{aligned}
\tag{5}
$$

where $\Omega$ represents the collection of all one-to-one mappings from $\{1, ..., n\}$ to $\{1, ..., n\}$. Now (5) is nuisance-free and can be used to estimate the unknown parameters in $p(v_i \mid z_i)$.

Although $\mathbf{V}$ is multi-dimensional in our case, the idea presented above can still be applied and it leads to

$$
\begin{aligned}
&\frac{\prod_{i=1}^{n} p(y_i, \mathbf{u}_i \mid z_i, r_i = 1)}{\Sigma_{\omega \in \Omega} \prod_{i=1}^{n} p(y_{\omega(i)}, \mathbf{u}_{\omega(i)} \mid z_i, r_i = 1)} \\
=\ & \frac{\prod_{i=1}^{n} p(y_i, \mathbf{u}_i \mid z_i)}{\Sigma_{\omega \in \Omega} \prod_{i=1}^{n} p(y_{\omega(i)}, \mathbf{u}_{\omega(i)} \mid z_i)}.
\end{aligned}
\tag{6}
$$

Furthermore, to simplify the computation, we adopt the pairwise fashion of (6) following the previous discussion in (Liang & Qin, 2000), which results

$$\prod_{1 \leq i < j \leq n} \frac{p(y_i, \mathbf{u}_i \mid z_i)p(y_j, \mathbf{u}_j \mid z_j)}{p(y_i, \mathbf{u}_i \mid z_i)p(y_j, \mathbf{u}_j \mid z_j) + p(y_i, \mathbf{u}_i \mid z_j)p(y_j, \mathbf{u}_j \mid z_i)}.$$

After plugging in the model (1) and some algebra, the

objective eventually becomes to minimize

$$L(\boldsymbol{\theta}) = \binom{N}{2}^{-1} \sum_{1 \le i < j \le N} \phi_{ij}(\boldsymbol{\theta}) \qquad (7)$$

$$= \binom{N}{2}^{-1} \sum_{1 \le i < j \le N} r_i r_j \log\{1 + W_{ij} \exp(\boldsymbol{\theta}^{\mathrm{T}} \mathbf{d}_{ij})\},$$

where $\boldsymbol{\theta} = (\widetilde{\gamma}, \widetilde{\boldsymbol{\beta}}^{\mathrm{T}})^{\mathrm{T}}$, $\widetilde{\gamma} = \gamma/\sigma^2$, $\widetilde{\boldsymbol{\beta}} = \widetilde{\gamma}\boldsymbol{\beta}$, $\mathbf{d}_{ij} = (-y_{i\backslash j} z_{i\backslash j}, \mathbf{u}_{i\backslash j}^{\mathrm{T}} z_{i\backslash j})^{\mathrm{T}}$, $y_{i\backslash j} = y_i - y_j$, $\mathbf{u}_{i\backslash j} = \mathbf{u}_i - \mathbf{u}_j$, $z_{i\backslash j} = z_i - z_j$ and

$$W_{ij} = \frac{p(z_i \mid \mathbf{u}_j)p(z_j \mid \mathbf{u}_i)}{p(z_i \mid \mathbf{u}_i)p(z_j \mid \mathbf{u}_j)}. \qquad (8)$$

Denote the minimizer of (7) as $\widehat{\boldsymbol{\theta}}$. By checking that $\frac{\partial^2 \phi_{ij}(\boldsymbol{\theta})}{\partial\boldsymbol{\theta}\partial\boldsymbol{\theta}^{\mathrm{T}}}$, equaling to

$$r_i r_j \{1 + W_{ij} \exp(\boldsymbol{\theta}^{\mathrm{T}} \mathbf{d}_{ij})\}^{-2} W_{ij} \exp(\boldsymbol{\theta}^{\mathrm{T}} \mathbf{d}_{ij}) \mathbf{d}_{ij} \mathbf{d}_{ij}^{\mathrm{T}},$$

is positive definite, the Hessian matrix is also positive definite, hence $\widehat{\boldsymbol{\theta}}$ uniquely exists. To compute $\widehat{\boldsymbol{\theta}}$, one also needs a model for $W_{ij}$. Fortunately, this model only depends on fully observed data $\mathbf{x}_i$ and $\mathbf{x}_j$. Essentially any existing parametric, semiparametric or nonparametric modeling techniques for $p(z \mid \mathbf{u})$ can be used, and $W_{ij}$ can be estimated accordingly. Throughout, we denote $\widehat{W}_{ij}$ as an available well-behaved estimator of $W_{ij}$.

One can notice that, due to the assumption (2) which allows the greatest flexibility of the mechanism model and the adoption of the conditional likelihood, not all parameters $\alpha$, $\boldsymbol{\beta}$, $\gamma$ and $\sigma^2$ are estimable. Nevertheless, the parameter $\boldsymbol{\beta}$, which quantifies the association between $Y$ and $\mathbf{U}$, is of primarily scientific interest and can be fully estimable. Our following paper focuses on the estimation and inference of $\boldsymbol{\beta}$, as well as the variable selection procedure based on $\boldsymbol{\beta}$.

In the past two decades, it has become a standard practice to consider the so-called high-dimensional setting where one usually assumes that $\boldsymbol{\beta}$ is sparse. While it is a prominent problem to consider the variable selection and post-selection inference when the data set is prone to missing values (Zhao et al., 2018; Yang et al., 2019; Jiang et al., 2020), the literature is quite scarce primarily because the treatment of missingness mechanism under high-dimensionality is cumbersome. In this paper, based on the idea presented above, we address the parameter estimation, variable selection and post-selection problems under model (1).

## 3. Computation

Directly minimizing $L(\boldsymbol{\theta})$ is feasible; however, it is very computationally challenging. From re-arranging the terms

in $L(\boldsymbol{\theta})$, we realize that it can be rewritten as the negative log-likelihood function of a standard logistic regression model. To be more specific, let $k$ be the index of pair $(i, j)$ with $k = 1, ..., K$ and $K = \binom{n}{2}$. Then

$$L(\boldsymbol{\theta}) = \frac{1}{K} \sum_{k=1}^{K} \log\left\{1 + \exp\left(s_k \boldsymbol{\theta}^{\mathrm{T}} \mathbf{t}_k + \log \widehat{W}_k\right)\right\}, \quad (9)$$

where $s_k = -\mathrm{sign}(z_{i\backslash j})$, $\mathbf{t}_k = (|z_{i\backslash j}| y_{i\backslash j}, -|z_{i\backslash j}| \mathbf{u}_{i\backslash j}^{\mathrm{T}})^{\mathrm{T}}$. Denote $g_k = I\{z_{i\backslash j} > 0\}$, then one can show that the summand in (9), $\log\left\{1 + \exp\left(s_k \boldsymbol{\theta}^{\mathrm{T}} \mathbf{t}_k + \log \widehat{W}_k\right)\right\}$, equals,

$$-\left[g_k\left(\boldsymbol{\theta}^{\mathrm{T}} \mathbf{t}_k + s_k \log \widehat{W}_k\right) - \log\left\{1 + \exp\left(\boldsymbol{\theta}^{\mathrm{T}} \mathbf{t}_k + s_k \log \widehat{W}_k\right)\right\}\right],$$

which is the contribution of the $k$-th subject to the negative log-likelihood function of a logistic regression with $g_k$ be the response, $\boldsymbol{\theta}$ be the coefficients, $\mathbf{t}_k$ be the covariate, $s_k \log \widehat{W}_k$ be the offset term, and there is no intercept. Therefore, $\widehat{\boldsymbol{\theta}}$ can be obtained by fitting the aforementioned logistic regression.

Regularization is a powerful technique to identify the zero elements of a sparse parameter in a regression model. Various penalty functions have been extensively studied such as LASSO (Tibshirani, 1996), SCAD (Fan & Li, 2001), and MCP (Zhang, 2010). In particular, we mainly study the adaptive LASSO penalty (Zou, 2006) with the objective of minimizing the following function

$$L_\lambda(\boldsymbol{\theta}) = L(\boldsymbol{\theta}) + \sum_{j=1}^{p} \lambda \left|\widehat{\widetilde{\beta}}_j\right|^{-1} \left|\widetilde{\beta}_j\right|, \qquad (10)$$

where $\lambda > 0$ is the tuning parameter. Following (Zou, 2006), $\widehat{\widetilde{\beta}}_j$ is a root-$N$-consistent estimator of $\widetilde{\beta}_j$; for example, one can use the estimator via minimizing the unregularized objective function (7). Obviously, adding a penalty term to $L(\boldsymbol{\theta})$ does not change its fundamental characteristic; $L_\lambda(\boldsymbol{\theta})$ is essentially the regularized log-likelihood of a logistic regression model with the similar features as discussed in (9).

The value of tuning parameter $\lambda$ is associated with the complexity of the selected model. The criteria to choose $\lambda$ has been extensively studied in the literature, such as the cross validation and various information-based criteria. In this paper, we follow the Bayesian information criterion (BIC) to determine $\lambda$. Specifically, we choose $\lambda$ to be the minimizer of the following BIC function

$$\mathrm{BIC}(\lambda) = 2L(\boldsymbol{\theta}) + p_\lambda \frac{\log(n)}{n}, \qquad (11)$$

where $p_\lambda$ is the number of nonzero elements in $\widehat{\widetilde{\boldsymbol{\beta}}}_\lambda$, the minimizer of (10).

## 4. Asymptotic Theory

In this Section, we first derive the asymptotic representation of $\widehat{\boldsymbol{\theta}}$, the minimizer of $L(\boldsymbol{\theta})$ without regularization, then that of $\widehat{\boldsymbol{\theta}}_\lambda$, the minimizer of our main objective function (10).

The asymptotic theory of $\widehat{\boldsymbol{\theta}}$ involves a model of $p(z \mid \mathbf{u})$, which does not contain any missing values hence any statistical model, either parametric, or semiparametric, or nonparametric, can be used. For simplicity, we only discuss the parametric case here, and any further elaborations can be similarly performed. For a parametric model $p(z \mid \mathbf{u}; \boldsymbol{\eta})$, one can apply the standard maximum likelihood estimate $\widehat{\boldsymbol{\eta}}$. Here, we simply assume

$$\sqrt{N}\left(\widehat{\boldsymbol{\eta}} - \boldsymbol{\eta}_0\right) \tag{12}$$

$$= -\mathbf{G}^{-1}\sqrt{N}\frac{1}{N}\sum_{i=1}^{N}\frac{\partial}{\partial \boldsymbol{\eta}}\log\left\{p(z_i \mid \mathbf{u}_i; \boldsymbol{\eta}_0)\right\} + o_p(1),$$

with $\mathbf{G} = E\left[\frac{\partial^2}{\partial\boldsymbol{\eta}\partial\boldsymbol{\eta}^{\mathrm{T}}}\log\left\{p(z \mid \mathbf{u}; \boldsymbol{\eta}_0)\right\}\right]$ and $E\|\frac{\partial^2}{\partial\boldsymbol{\eta}\partial\boldsymbol{\eta}^{\mathrm{T}}}\log\left\{p(z \mid \mathbf{u}; \boldsymbol{\eta}_0)\right\}\|^2 < \infty$. With this prerequisite, we have the following result for $\widehat{\boldsymbol{\theta}}$.

**Theorem 1.** *Assume* $E\left\|\frac{\partial^2\phi_{ij}(\boldsymbol{\theta}_0,\boldsymbol{\eta}_0)}{\partial\boldsymbol{\theta}\partial\boldsymbol{\theta}^{\mathrm{T}}}\right\|^2 < \infty$. *Then* $\sqrt{N}\left(\widehat{\boldsymbol{\theta}} - \boldsymbol{\theta}_0\right) \xrightarrow{d} N\left(\mathbf{0}, \mathbf{A}^{-1}\boldsymbol{\Sigma}\mathbf{A}^{-1}\right)$, *where* $\mathbf{A} = E\left\{\frac{\partial^2\phi_{ij}(\boldsymbol{\theta}_0,\boldsymbol{\eta}_0)}{\partial\boldsymbol{\theta}\partial\boldsymbol{\theta}^{\mathrm{T}}}\right\}$, $\boldsymbol{\Sigma} = 4E\left\{\boldsymbol{\lambda}_{12}(\boldsymbol{\theta}_0,\boldsymbol{\eta}_0)\boldsymbol{\lambda}_{13}(\boldsymbol{\theta}_0,\boldsymbol{\eta}_0)^{\mathrm{T}}\right\}$, $\boldsymbol{\lambda}_{ij}(\boldsymbol{\theta}_0,\boldsymbol{\eta}_0) = \mathbf{B}\mathbf{G}^{-1}\mathbf{M}_{ij}(\boldsymbol{\eta}_0) - \mathbf{N}_{ij}(\boldsymbol{\theta}_0,\boldsymbol{\eta}_0)$, $\mathbf{B} = E\left\{\frac{\partial^2\phi_{ij}(\boldsymbol{\theta}_0,\boldsymbol{\eta}_0)}{\partial\boldsymbol{\theta}\partial\boldsymbol{\eta}^{\mathrm{T}}}\right\}$, $\mathbf{M}_{ij}(\boldsymbol{\eta}_0) = \frac{1}{2}\left\{\frac{\partial}{\partial\boldsymbol{\eta}}\log p(z_i \mid \mathbf{u}_i; \boldsymbol{\eta}_0) + \frac{\partial}{\partial\boldsymbol{\eta}}\log p(z_j \mid \mathbf{u}_j; \boldsymbol{\eta}_0)\right\}$, *and* $\mathbf{N}_{ij}(\boldsymbol{\theta}_0,\boldsymbol{\eta}_0) = \frac{\partial\phi_{ij}(\boldsymbol{\theta}_0,\boldsymbol{\eta}_0)}{\partial\boldsymbol{\theta}}$.

Recall that $\boldsymbol{\theta} = (\widetilde{\gamma}, \widetilde{\boldsymbol{\beta}}^{\mathrm{T}})^{\mathrm{T}}$. Without loss of generality, we assume that the first $p_0$ parameters in $\widetilde{\boldsymbol{\beta}}$ are nonzero, where $1 \leq p_0 < p$. For simplicity, we denote $\boldsymbol{\theta}_T = (\widetilde{\gamma}, \widetilde{\beta}_1, ..., \widetilde{\beta}_{p_0})^{\mathrm{T}}$ as the vector of nonzero components and $\boldsymbol{\theta}_{T^C} = (\widetilde{\beta}_{p_0+1}, ..., \widetilde{\beta}_p)^{\mathrm{T}}$ as the vector of zeros. Let $\mathbf{A} = \frac{\partial^2\phi_{ij}(\boldsymbol{\theta}_0,\boldsymbol{\eta}_0)}{\partial\boldsymbol{\theta}\partial\boldsymbol{\theta}^{\mathrm{T}}}$, a $(p+1) \times (p+1)$ matrix, can be partitioned as $\mathbf{A} = \begin{pmatrix} \mathbf{A}_1 & \mathbf{A}_2 \\ \mathbf{A}_2^{\mathrm{T}} & \mathbf{A}_3 \end{pmatrix}$, where $\mathbf{A}_1$ is a $(p_0 + 1) \times (p_0 + 1)$ submatrix corresponding to $\boldsymbol{\theta}_T$. Recall that in Theorem 1, we define $\boldsymbol{\Sigma} = 4E\left\{\boldsymbol{\lambda}_{12}(\boldsymbol{\theta}_0,\boldsymbol{\eta}_0)\boldsymbol{\lambda}_{13}(\boldsymbol{\theta}_0,\boldsymbol{\eta}_0)^{\mathrm{T}}\right\}$, also a $(p+1) \times (p+1)$ matrix. Now we assume it can be partitioned as $\boldsymbol{\Sigma} = \begin{pmatrix} \boldsymbol{\Sigma}_1 & \boldsymbol{\Sigma}_2 \\ \boldsymbol{\Sigma}_2^{\mathrm{T}} & \boldsymbol{\Sigma}_3 \end{pmatrix}$, where $\boldsymbol{\Sigma}_1$ is a $(p_0 + 1) \times (p_0 + 1)$ submatrix corresponding to $\boldsymbol{\theta}_T$ as well. Then we present the asymptotic normality for the nonzero components as well as the consistency in variable selection, the so-called oracle property of $\widehat{\boldsymbol{\theta}}_\lambda$.

**Theorem 2.** *Assume* $\boldsymbol{\theta}_0$ *exists and is unique,* $\mathbf{A}_1$ *is positive definite,* $E\|\frac{\partial\phi_{ij}(\boldsymbol{\theta}_0,\boldsymbol{\eta}_0)}{\partial\boldsymbol{\theta}}\|^2 < \infty$ *for each* $\boldsymbol{\theta}$ *in a neighbourhood of* $\boldsymbol{\theta}_0$. *In addition, we assume* $\sqrt{N}\lambda \to 0$ *and* $N\lambda \to \infty$. *Then*

$$\sqrt{N}\left(\widehat{\boldsymbol{\theta}}_{\lambda,T} - \boldsymbol{\theta}_{0,T}\right) \xrightarrow{d} N\left(\mathbf{0}, \mathbf{A}_1^{-1}\boldsymbol{\Sigma}_1\mathbf{A}_1^{-1}\right).$$

*In addition, let* $T_N = \{j \in \{1, ..., p\} : \widehat{\widetilde{\beta}}_{j,\lambda} \neq 0\}$ *and* $T = \{j \in \{1, ..., p\} : \widetilde{\beta}_{j,0} \neq 0\}$, *then*

$$\lim_{N\to\infty} pr(T_N = T) = 1.$$

## 5. Variance Estimation

Although the above theory provides a rigorous justification for the asymptotic property of $\widehat{\boldsymbol{\theta}}_\lambda$, in practice, however, it does not guide the standard error estimation. Here we propose a data perturbation approach for the variance estimation. Specifically, following (Cai et al., 2005), we generate a set of independent and identically distributed positive random variables $\boldsymbol{\Xi} = \{\xi_i, i = 1, ..., N\}$ with $E(\xi_i) = 1$ and $\text{var}(\xi_i) = 1$, e.g., the standard exponential distribution. Since it is based on a U-statistic structure, we perturb our objective function by adding $\kappa_{ij} = \xi_i\xi_j$ to each of its pairwise term. We first obtain the estimator $\widehat{\boldsymbol{\theta}}^\star$ by minimizing the perturbed version of (7), $L^\star(\boldsymbol{\theta}) = \binom{N}{2}^{-1}\sum_{1\leq i < j \leq N}\kappa_{ij}\phi_{ij}(\boldsymbol{\theta})$. Then we obtain the estimator $\widehat{\boldsymbol{\theta}}_\lambda^\star$ by minimizing the perturbed version of (10):

$$L_\lambda^\star(\boldsymbol{\theta}) = \binom{N}{2}^{-1}\sum_{1\leq i < j \leq N}\kappa_{ij}\phi_{ij}(\boldsymbol{\theta}) + \sum_{j=1}^{p}\frac{\lambda}{\left|\widehat{\widetilde{\beta}}_j^\star\right|}\left|\widetilde{\beta}_j\right|,$$

where the optimal $\lambda$ is also computed by the BIC.

Following the theory of (Kosorok, 2007) and (Minnier et al., 2011), under some regularity conditions, one can first show that $\sqrt{N}\left(\widehat{\boldsymbol{\theta}}_{\lambda,T}^\star - \boldsymbol{\theta}_{0,T}\right)$ converges in distribution to $N(\mathbf{0}, \mathbf{A}_1^{-1}\boldsymbol{\Sigma}_1\mathbf{A}_1^{-1})$, the same limiting distribution of $\sqrt{N}\left(\widehat{\boldsymbol{\theta}}_\lambda - \boldsymbol{\theta}_0\right)$. Furthermore, $\text{pr}^*\left(\widehat{\boldsymbol{\theta}}_{\lambda,T^C}^\star = 0\right) \to 1$, where $\text{pr}^*$ is the probability measure generated by the original data we have $\mathcal{X}$ and the perturbation data $\boldsymbol{\Xi}$. In addition, one can show that the distribution of $\sqrt{N}\left(\widehat{\boldsymbol{\theta}}_{\lambda,T}^\star - \widehat{\boldsymbol{\theta}}_{\lambda,T}\right)$ conditional on the data can be used to approximate the unconditional distribution of $\sqrt{N}\left(\widehat{\boldsymbol{\theta}}_{\lambda,T} - \boldsymbol{\theta}_{0,T}\right)$ and that $\text{pr}^*\left(\widehat{\boldsymbol{\theta}}_{\lambda,T^C}^\star = 0 \mid \mathcal{X}\right) \to 1$.

To achieve a confidence region for $\theta_j$, the $j$-th coordinate in $\boldsymbol{\theta}$, the lower and upper bounds can be formed by $\widehat{\theta}_{\lambda,j,\alpha/2}^\star$ and $\widehat{\theta}_{\lambda,j,1-\alpha/2}^\star$ respectively, where $\widehat{\theta}_{\lambda,j,q}^\star$ represents the $q$-th quantile of $\left\{\widehat{\theta}_{\lambda,j,m}^\star, m = 1, ..., M\right\}$.

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
