# OpenReview forum: "A Penalized Likelihood Approach for Statistical Inference in a High-Dimensional Linear Model with Missing Data"
_ICML.cc/2020/Workshop/Artemiss — ICML Artemiss 2020_

### Official Review · AnonReviewer2 · 2020-06-24
**A Penalized Likelihood Approach for Statistical Inference in a High-Dimensional Linear Model with Missing Data**

**Confidence:** 5
**Rating:** 7

**Review:**

The paper deals with high-dimensional linear regression, in the presence of missing values in the response. The authors propose a semi-parametric framework which allows general missing data mechanisms (MCAR, MAR, MNAR), through the introduction of a shadow variable conditionnally independent of the missing data pattern. They also compute the conditional likelihood, and propose an algorithm to minimize it. Finally, an important contribution is the proof of consistency and asymptotic normality of the resulting estimator.

---

### Decision · Program_Chairs · 2020-07-02

**Decision:**

Accept

**Comment:**

We're happy to accept this paper at Artemiss. We'll contact you soon to inform you about more details concerning the format of your presentation at the workshop, and the camera-ready version deadline. Please take into account the referee's comments to write the camera-ready version.